# Functional *ERAP1* Variants Distinctively Associate with Ankylosing Spondylitis Susceptibility under the Influence of *HLA-B27* in Taiwanese

**DOI:** 10.3390/cells11152427

**Published:** 2022-08-05

**Authors:** Chin-Man Wang, Ming-Kun Liu, Yeong-Jian Jan Wu, Jing-Chi Lin, Jian-Wen Zheng, Jianming Wu, Ji-Yih Chen

**Affiliations:** 1Department of Rehabilitation, Chang Gung Memorial Hospital, Chang Gung University College of Medicine, Taoyuan City 333, Taiwan; 2Department of Medicine, Division of Allergy, Immunology and Rheumatology, Chang Gung Memorial Hospital, Chang Gung University College of Medicine, Taoyuan City 333, Taiwan; 3Department of Veterinary and Biomedical Sciences, Department of Medicine, University of Minnesota, St. Paul, MN 55108, USA

**Keywords:** ankylosing spondylitis, ERAP1, single nucleotide variant, HLA-B27, HLA class I molecules

## Abstract

Epistasis of *ERAP1* single nucleotide variations (SNVs) and *HLA-B27* has been linked to ankylosing spondylitis susceptibility (AS). The current study examined how prevalent *ERAP1* allelic variants (SNV haplotypes) in Taiwan affect ERAP1 functions and AS susceptibility in the presence or absence of HLA-B27. Sanger sequencing was used to discover all *ERAP1* coding SNVs and common allelic variants in Taiwanese full-length cDNAs from 45 human patients. For the genetic association investigation, TaqMan genotyping assays were utilized to establish the genotypes of *ERAP1* SNVs in 863 AS patients and 1438 healthy controls. Ex vivo biological analysis of peripheral blood mononuclear cells from homozygous donors of two common-risk *ERAP1* allelic variants was performed. Two common-risk *ERAP1* allelic variants were also cloned and functionally studied. In Taiwanese, eleven frequent *ERAP1* SNVs and six major *ERAP1* allelic variants were discovered. We discovered that in Taiwanese, the most prevalent *ERAP1*-001 variant with 56E, 127R, 276I, 349M, 528K, 575D, 725R, and 730Q interacting with *HLA-B27* significantly contributed to the development of AS. In *HLA-B27* negative group, however, the second most prevalent *ERAP1*-002 variant with 56E, 127P, 276M, 349M, 528R, 575D, 725R, and 730E was substantially related with an increased risk of AS. Ex vivo and in vitro research demonstrated that *ERAP1* allelic variants have a significant impact on ERAP1 functions, suggesting that ERAP1 plays a role in the development of AS. In an HLA-B27-dependent manner, common *ERAP1* allelic variants are related with AS susceptibility.

## 1. Introduction

Ankylosing spondylitis (AS) is a subtype of axial spondylopathy (AxSpA) that primarily affects the axial skeleton and is characterized by the formation of syndesmophytes and spinal ankylosis deformities [1]. The pathophysiology of AS, which has long been thought to be a highly familial and heritable disease [2,3], is complicated by a number of variables. Multiple genes may be involved in the development of AS, based on the heterogeneity of the disease’s progression. Several genes and genomic areas have been linked to AS susceptibility and severity in genome-wide association studies (GWAS), demonstrating that both MHC and non-MHC genes have a role in the illness process [4,5,6,7,8]. Despite breakthroughs in recent years, the genetic and pathophysiology processes of AS remain poorly known.

Antigenic peptides on MHC molecules are scanned by the human immune system to discern infected or sick cells. Antigenic peptides degraded in proteasomes are then trimmed to appropriate lengths on the ER luminal side by aminopeptidases for correct presentation on MHC-I molecules [9,10,11,12,13]. Endoplasmic reticulum aminopeptidase 1 (ERAP1), a unique member of the M1 aminopeptidase family, trims antigen peptides to fit the antigen binding site on MHC-I during the final stage of antigen processing. Peptide–MHC-I complexes on the cell surface reflect the existing proteome of respective cells.

The function of ERAP1 has a major impact on the stability and immunological characteristics of MHC-I molecules because of its critical involvement in processing MHC-I ligands. Low or excessive ERAP1 activity could result in antigen peptides that are unsuitable for MHC-I presentation. As a result, effective ERAP1 function is required for optimal antigen presentation and health maintenance. *ERAP1* is a non-MHC gene on chromosome 5q15, and genetic studies have shown that epistasis between ERAP1 SNVs and HLA-B27 is highly linked to AS vulnerability in different ethnic groups [6,14]. The biological functions of *ERAP1* allelic variants on the other hand, are largely unexplored. The molecular processes underpinning the link between *ERAP1* allelic variations and AS are still being investigated [15]. Furthermore, different combinations of *ERAP1* SNVs (or allelic variants) may interact with *HLA-B* in a unique way to contribute to AS vulnerability in a single ethnic population. The breadth of *ERAP1* SNVs and allelic variations in Taiwanese was investigated in this study. The *ERAP1*-001 variation was discovered to be a key risk factor for AS in Taiwanese people. Furthermore, in an *HLA-B27*-dependent manner, two prevalent *ERAP1* allelic variants are strongly related with AS susceptibility.

## 2. Materials and Methods

### 2.1. Study Subjects

A total of 863 AS patients (717 males and 146 females; onset ages ranged from 8 to 72 with the mean onset age of 25.38-year-old) who fulfilled the 1984 revised New York diagnostic criteria for AS were recruited at the Chang Gung Memorial Hospital. HLA-B27 carriers accounted for 801 (92.8%) of the 863 AS patients. As normal healthy controls, a total of 1438 adult healthy blood donors (706 males and 732 females; ages varied from 18 to 65, with the mean age of 40.78-year-old) from the same geographical region were recruited. All human research protocols were approved by the ethics committee of Chang Gung Memorial Hospital, and all subjects gave their informed consent.

### 2.2. HLA-B27 Determination

Flow cytometry and/or PCR tests were used to determine HLA-B27 positive. Whole blood samples were analyzed by flow cytometry for HLA-B27 using the BDTM HLA-B27 kit (BD Biosciences, Franklin Lakes, NJ, USA), which combines fluorescein-conjugated anti-HLA-B27 and phycoerythrin-conjugated anti-CD3 antibodies.

### 2.3. HLA-B Allele Determination

On ABI 3730XL DNA Analyzers, HLA-B alleles were determined using a commercial sequencing-based typing (SBT) kit (TBG Biotechnology Corp., New Taipei City, Taiwan). The AccuType HLA SBT Analysis Software assigned HLA-B alleles (TBG Biotechnology Corp, New Taipei City, Taiwan).

### 2.4. Genomic DNA Isolation

As previously disclosed [16], genomic DNA was recovered from EDTA anticoagulated peripheral blood using the Puregene DNA isolation kit (Gentra Systems, Minneapolis, MN, USA).

### 2.5. RNA Isolation and cDNA Synthesis

TRIzol total RNA isolation reagent was used to isolate total RNA from human peripheral blood leukocytes (Invitrogen, Carlsbad, CA, USA). For deep sequencing investigation of ERAP1 coding SNVs and creation of ERAP1 expression constructs, the SuperScript Preamplification system (Invitrogen, Carlsbad, CA, USA) was utilized to synthesize peripheral blood leukocyte cDNA.

### 2.6. Deep DNA Sequencing Analyses of ERAP1 SNVs in Taiwanese

To identify Taiwanese-specific *ERAP1* SNVs for the population genetic study, we carried out RT-PCR to amplify the full-length *ERAP1* cDNA (2949 bps) using the sense primer (5′-CCC AGA ACC CCA GGT AGG TA-3′) and the antisense primer (5′-GCT ATG CTT CCA TTC CGT TT-3′). The *ERAP1* cDNA sequence reactions were performed using the BigDye Terminator Sequencing kit (Applied Biosystem, Foster City, CA, USA) and the following seven sequencing primers: (1) 5′-AAG TCT CCG AAA GAT TGC CAG CAT-3′, (2) 5′-GGT TTC TGT TTA TGC TGT GCC AGA C-3′, (3) 5′-GAT CTT GTT TGG GTA GGG GAT ACG G-3′, (4) 5′-GCA AGA GCA CTA CAT GAA GGG CTC-3′, (5) 5′-TCA TGC CCA CAT TAA ATT TGA TCC A-3′, (6) 5′-AGG AAA AGC TTC AAT GGC TAC TAG A-3′, and (7) 5′-CAA TTG TCT GTT GGA CAC AAC GGA-3′. Sequencing tracer data were analyzed using SeqMan Pro software (DNAstar Inc., Madison, WI, USA) to identify and verify *ERAP1* SNVs.

### 2.7. ERPA1 SNV Genotype Analyses

The common *ERPA1* SNVs identified in Taiwanese population were genotyped using made-to-order allelic discrimination assays from Applied Biosystems (ABI, Foster City, CA, USA). SNV genotyping assays were carried out on a vi7 real-time PCR system (ABI). The SNV genotypes were determined using ABI TaqMan Genotyper software (Foster City, CA, USA) according to the vendor’s instructions.

### 2.8. ERAP1 Expression Constructs

The *Xho* I/*Xba* I-flanked Lir-EGFP DNA fragment from pBacCMV-MCS-Lir-EGFP was amplified by PCR using the forward primer 5′-TTCTCGAGTTAAAACAGCCTGTGGGT-3′ (the *Xho* I site is underlined) and the reverse primer 5′-GGCTCTAGATTACTTGTACAGCTCGTCCA-3′ (the *Xba* I site is underlined) and the fragment was subsequently cloned into pcDNA3.1 vector (Invitrogen) at *Xho* I and *Xba* I restriction enzyme sites for the generation of pcDNA3.1-Lir-EGFP vector. The full-length *ERAP1* cDNAs from *ERAP1*-001 homozygous donors and *ERAP1*-002 homozygous donors were amplified using the forward primer 5′-ACGGGAATTCCCACCATGGTGTTTCTGCCCCTCAA-3′ (the *Eco*R I site is underlined) and the reverse primer 5′-AACGGGCTCGAGTCATCCTGTTGCGTCAGCTTCA-3′ (the *Xho* I site is underlined). The *ERAP1* cDNA inserts were cloned into pcDNA3.1-Lir-EGFP vector digested with *Eco*R I and *Xho* I restriction enzymes.

### 2.9. Peripheral Blood Mononuclear Cell (PBMC) Isolation and Culture

Heparin collection tubes were used to collect whole blood from donors. Centrifugation through FicollPaque Premium (GE Healthcare, Chicago, IL, USA) density gradients was used to isolate PBMCs. PBMCs were cultured for 48 h in RPMI-1640 with 10% FBS and 10 ng/mL phorbol 12-myristate 13-acetate (PMA) before being stimulated for 16 h with LPS (1 µg/mL). The supernatants of PBMC cultures were then collected for IL-23 analysis, and cells were extracted for Western blot analysis.

### 2.10. Cell Culture

CHO (Chinese hamster ovary cells) cells were grown in DMEM with 10% fetal bovine serum (Sigma, St Louis, MO, USA). U937 (human monocytic leukemia) cells were grown in RPMI 1640 medium, which included 2 mM L-glutamine, 4.5 g/L glucose, 10 mM HEPES, 1 mM sodium pyruvate, 100 U/mL penicillin/streptomycin, and 10% fetal bovine serum. All cells were kept at 37 °C in a 5% CO_2_ environment.

### 2.11. Transfection of CHO and U937 Cells

*Trans*IT transfection reagent (Mirus, Madison, WI, USA) was used for transfection of *ERAP1* expression constructs into cells. On 6-well plates, 2 × 10^6^ cells were seeded in each well. Before transfection, the cells were washed twice with medium devoid of serum. *ERAP1* expression construct plasmid DNA (2.5 g) was diluted in 400 µL serum-free DMEM medium and then combined with *Trans*IT transfection reagent (5 µL) to generate DNA-*Trans*IT complexes. The DNA-*Trans*IT complexes were added to cells in 400 µL of serum-free media per well. After 5 h of incubation, the medium was discarded and replaced with 2 mL of new medium containing 10% fetal bovine serum. After 24 h of transfection, the efficacy of transfection was evaluated by fluorescence microscopy.

### 2.12. Western Blot Analysis

On a micro Protein III system, cell lysates were subjected to SDS–PAGE for protein separation (Bio-Rad, Hercules, CA, USA). Proteins on the gel were electrotransferred onto a polyvinylidene difluoride (PVDF) membrane following SDS–PAGE. Tris-buffered saline (TTBS; 100 mmol/L Tris (pH 7.4), 100 mmol/L NaCl, and 0.1 percent Tween 20) containing 5% (*v*/*v*) non-fat milk was used to block PVDF membrane at room temperature for 1 h with moderate shaking. The membrane was then shaken overnight at 4 °C with primary antibody diluted in TBS containing 0.5 percent (*v*/*v*) nonfat milk. The membrane was washed three times for ten minutes at room temperature with TBS solution to remove unbound antibodies. The membrane was then treated at room temperature for one hour with secondary antibodies coupled with horseradish peroxidase (HRP). Enhanced Chemiluminescence kit (Pierce, Rockford, IL, USA) and the imaging equipment were used to detect the HRP-labeled proteins on membrane (UVP ChemStudio PLUS Touch, Analytik Jena US LLC, Cambridge, UK). Band intensities were quantified from the digital image by densitometry using ImageJ software (NIH, Bethesda, MD, USA).

### 2.13. ERAP1 Protein Immunoprecipitation

Cell pellets (25 × 10^6^ cells for lymphoblastoid B cells or 2 × 10^6^ cells for PBMCs) were lysed on ice for 30 min in lysis buffer (M-PER™ Mammalian Protein Extraction Reagent, Thermo Scientific, Waltham, MA, USA). The lysate supernatants were extracted by centrifuging at 14,000 rpm for 10 min at 4 degrees Celsius. Using the BCA (bicinchoninic acid) technique, protein concentrations were determined prior to immunoprecipitation. The lysates were then precleared by incubation with glycine Sepharose beads prior to treatment with anti-ERAP1 4D2-mAb-conjugated Sepharose beads for at least 2 h. The beads were subsequently washed three times in buffer (50 mM Tris, 150 mM NaCl, pH 7.4) and resuspended in buffer (500 µL) prior to use in the activity experiment.

### 2.14. ERAP1 Enzyme Activity Determination

The effect of *ERAP1* allelic variants on ERAP1 enzyme activity was analyzed using a continuous fluorigenic assay as described [17]. Enzymatic reactions were carried out at 37 °C. In all reactions, the substrate concentration exceeded the enzyme concentration by at least 100-fold. Typically, 10 to 1000 nM ERAP1 protein was combined with 1 to 100 µM peptide, and the fluorescence of the reaction was monitored. On a QuantaMaster 4 spectrofluorimeter (Photon Technology International, Birmingham, NJ, USA) and a Tecan SpectraFluor plate reader, fluorescence measurements were conducted. Excitation at 295 nm and emission at 350 nm were used to acquire fluorescence spectra.

### 2.15. Cytokine Determination

ELISA kits were obtained from BioLegend (San Diego, CA, USA). All assays were conducted according to the manufacturer’s instructions. Absorbencies were measured using MAXlineMicroplate Readers at 450 nm (Molecular devices, San Jose, CA, USA). The data are presented as the mean ± standard deviation of technical replicates.

### 2.16. Statistical Analysis

Linear regression analysis and Pearson’s coefficient of correlation were used to analyze the correlation between different parameters. The distributions of SNV alleles and genotypes in patients and controls were analyzed using single-locus analysis. These comparisons were made using three chi-square tests (the genotype case-control test, the allele case-control test, and the Cochran-Armitage trend test). Using the SAS/Genetics software package release 8.2 (SAS Institute, Cary, NC, USA), significant relationships of SNVs with characteristics (*p* < 0.05) were detected, and *p*-values, odds ratios (ORs), and 95 percent confidence intervals (CIs) were calculated. The reported probability of false-discovery rate (PFDR) was estimated using Q-VALUE software (http://genomics.princeton.edu/storeylab/qvalue accessed on 6 June 2022) to account for multiple simultaneous statistical comparisons. Haploview 4.2 (Broad Institute, Cambridge, MA, USA; http://www.broad.mit.edu/mpg/haploview accessed on 6 June 2022) was used to calculate linkage disequilibrium (LD) between marker loci and to create haplotype blocks. We employed illness status (case vs. control) and HLA-B27 positivity to test for haplotype-trait association for markers within the same haplotype block (or allelic variations). To study genetic correlations, researchers utilized stepwise logistic regression analysis to account for sex. For all analyses, a 5% level of significance was employed for *p*-values.

## 3. Results

### 3.1. Associations of ERAP1 SNVs with AS Susceptibility

By sequencing the full-length *ERAP1* cDNAs from 45 Taiwanese subjects randomly selected for SNV discovery purpose, we identified nine common *ERAP1* coding SNVs (cSNVs) including eight non-synonymous cSNVs that cause amino acid substitutions (Appendix A). We subsequently carried out genotype analyses of nine *ERAP1* cSNVs plus two *ERAP1* intron SNVs that were previously reported to associate with AS susceptibility [18,19]. Four *ERAP1* cSNVs (rs26653G > C [R127P], rs30187A > G [K528R], rs469783G > A [A637A], and rs27044C > G [Q730E]) and two intron SNVs (rs27980A > C and rs27037T > G) had significantly different genotype and allele distributions between 863 AS patients and 1438 healthy controls (Appendix A). The cSNV alleles consisting of rs26653G or 127R (Trend test *P_FDR_* (*TtP**_FDR_*)= 0.0024; additive model adjusted for sex *P_FDR_* (*add_sex_ P_FDR_*) *=* 0.0689, OR 1.17 [95% CI 1.01–1.35]), rs30187A (528K) (*TtP**_FDR_* < 0.00001; *add_sex_ P_FDR_* = 0.034, OR 1.20 [95% CI 1.04–1.38]), rs469783G (silent SNP) (*TtP**_FDR_* < 0.00001; *add_sex_ P_FDR_* = 0.0029, OR 1.28 [95% CI 1.11–1.48]), and rs27044C (730Q) (*TtP**_FDR_* < 0.00001; *add_sex_ P_FDR_* = 0.01, OR 1.25 [95% CI 1.08–1.44]), intron SNV rs27980A (*TtP**_FDR_* = 0.0097; *add_sex_ P_FDR_* = 0.0446, OR 1.17 [95% CI 1.03–1.34]), and the intron rs27037T (*TtP**_FDR_* < 0.00001; *add_sex_ P_FDR_* = 0.0029, OR 1.30 [95% CI 1.12–1.50]) alleles were similarly substantially related with AS susceptibility (Appendix A).

### 3.2. Association of ERAP1 SNVs with HLA-B27 Positivity among AS Patients

Because *HLA-B27* plays a causal role in the development of AS, we investigated the association between *ERAP1* SNVs and *HLA-B27* status in AS patients. As shown in Appendix A, the allele distributions of six *ERAP1* SNVs (rs26653, rs26618, rs30187, rs469783, rs27044, and rs27037) differed significantly between *HLA-B27* positive patients (*HLA-B27*^+^) and the *HLA-B27* negative patients (*HLA-B27*^−^). Five cSNV alleles are rs26653G (*TtP**_FDR_* = 0.0083; *add_sex_ P_FDR_* = 0.0115, OR 1.73 [95% CI 1.19–2.52]), rs26618A (*TtP**_FDR_* = 0.0044; *add_sex_ P_FDR_* = 0.0021, OR 2.12 [95% CI 1.43–3.14]), rs30187A (*TtP**_FDR_* = 0.0083; *add_sex_ P_FDR_* = 0.0099, OR 1.74 [95% CI 1.21–2.51]), rs469783G (*TtP**_FDR_* = 0.0099; *add_sex_ P_FDR_* = 0.0126, OR 1.69 [95% CI 1.16–2.440]), and rs27044G (*TtP**_FDR_* = 0.0077; *add_sex_ P_FDR_* = 0.0094, OR 1.80 [95% CI 1.25–2.60]). The intron SNV rs27037T allele was also substantially related with HLA-B27 positive in AS patients (*TtP**_FDR_* = 0.0369; *add_sex_ P_FDR_* = 0.0270, OR 1.60 [95% CI 1.10–2.34]).

### 3.3. Association of ERAP1 Allelic Variants (SNV Haplotypes) with AS Susceptibility

Linkage disequilibrium (LD) block analysis reveals that the majority of 11 *ERAP1* SNVs are in strong linkage disequilibrium (Appendix A). Taiwanese haplotype analysis found that eleven *ERAP1* SNVs included six main allelic variations (haplotypes) (Appendix A). The *ERAP1*-001 comprised of rs3734016G-rs26653G-rs26618A-rs2287987A-rs30187A-rs10050860G-rs469783G-rs10050860G-rs469783G-s17482078G-rs27044C-rs27980A-rs27037T is the most prevalent allelic variant in Taiwanese healthy population (Table 1). Five cSNV sites (rs26653G > C [R127P], rs26618A > G [I276M], rs30187A > G [K528R], rs469783G > A [A637A], and rs27044C > G [Q730E]) and two intron SNP sites (rs27980 and rs27037) distinguish *ERAP1*-001 from *ERAP1*-002 (Appendix A). *ERAP1*-001 was significantly enriched in AS patients (frequency = 0.4685) compared to healthy controls (frequency = 0.3669) (Adjusted *p* = 1.39E-011, OR = 1.53 [95% CI 1.35–1.73]), whereas the frequencies of the other five *ERAP1* variants did not differ significantly between AS patients and healthy controls (Table 1). According to our findings, the *ERAP1*-001 variant is a significant risk factor for AS in Taiwanese.

### 3.4. ERAP1 Allelic Variants Interact with HLA-B27 to Affect AS Susceptibility

*HLA-B27* is a well-established AS risk factor. Following this, we studied the association between *ERAP1* SNV haplotypes and *HLA-B27* status in AS patients stratified by *HLA-B27* positivity. The allele frequencies of *ERAP1*-001, -002, and -004 differed considerably between *HLA-B27*^+^ AS patients and H *HLA-B27*^−^ AS patients, as indicated in Table 2. The *ERAP1*-001 frequency was significantly greater (Adjusted *p* = 0.01472, OR = 1.61 [95% CI: 1.10–2.36]) in *HLA-B27*^+^ AS patients (frequency = 0.4777) than in *HLA-B27*^−^ AS patients (frequency = 0.3680) (Table 2). Intriguingly, the frequency of the rare ERAP1-004, which differs from the *ERAP1*-001 by only two intron SNVs (Appendix A), was significantly higher (Adjusted *p* = 0.0409, OR 2.90 [95% CI 1.04–8.04]) in *HLA-B27*^+^ AS patients (frequency = 0.0818) than in *HLA-B27*^−^ AS patients (frequency = 0.0310) (Table 2). Notably, the *ERAP1*-002 was substantially higher (Adjusted *p* = 0.00064, OR 0.50 [95% CI 0.34–0.75]) in *HLA-B27*^−^ AS patients (frequency = 0.3395) than in *HLA-B27*^+^ AS patients (frequency = 0.2089), showing that *ERAP1*-002 is a risk factor for AS in HLA-B27^−^ persons. The *ERAP1*-001 allelic frequency tended to be higher in *HLA-B27*^+^ AS patients (frequency = 0.4768) than in *HLA-B27*^+^ healthy controls (frequency = 0.4021); however, the difference was not statistically significant, most likely due to the limited sample size of *HLA-B27*+ healthy controls (Appendix A). In contrast, the *ERAP1*-002 frequency was significantly higher (Adjusted *p* = 0.008, OR 1.69 [95% CI 1.15–2.49]) in *HLA-B27*^−^ AS patients (frequency = 0.3453) than in *HLA-B27*^−^ healthy controls (frequency = 0.2365) (Table 3), indicating that *ERAP1*-002 is a major risk factor for AS in HLA-B27^−^ individuals. *ERAP1*-001 and -004 are risk factors for AS in *HLA-B27*^+^ persons, but *ERAP1*-002 is a risk factor for AS in HLA-B27^−^ individuals. Our findings support the hypothesis that *ERAP1* variants interact with *HLA-B* to differentially influence Taiwanese susceptibility to AS.

### 3.5. ERAP1 Variant Genotypes Affect Phenotypes

Taiwanese ERAP1 variations have a number of amino acid changes (Appendix A). To examine the epistaxis effects if ERAP1 haplotypes influence phenotypes in HLA-B27 positive individuals, peripheral blood mononuclear cells (PBMCs) were extracted from 11 human participants (seven *ERAP1*-001 homozygous AS patients and four *ERAP1*-002 AS patients) having a single HLA-B2704 allele. As demonstrated in Figure 1, PBMCs from *ERAP1*-001 homozygous donors produced more ERAP1 protein, MHC class I free heavy chain (FHCs) and FHC dimers, unfolded protein response (UPR) markers (immunoglobulin heavy-chain binding protein (BiP), CCAAT-enhancer-binding protein homologous protein (CHOP) and X-box-binding protein 1 (XBP1), autophagy markers (Beclin-1, LC3 I and LC3 II), and inflammation markers (Caspase 1 and IL-1β) as compared to those from *ERAP1*-002 homozygous donors. Notably, PBMCs of seven *ERAP1*-001 homozygous donors produced significantly higher levels of IL-23 p19 and p40 than PBMCs from four *ERAP1*-002 homozygous donors.

Previous studies indicates that non-conservative *ERAP1* SNVs, such as K528R and R725Q substitutions, may impact ERAP1 protein production and aminopeptidase activity [14]. We conducted in vitro experiments to evaluate the effects of *ERAP1* variations on ERAP1 expression and aminopeptidase activity. The variant cDNAs of *ERAP1*-001 and *ERAP1*-002 were cloned into the bi-cistronic mammalian expression vector pcDNA-Lir-EGFP. Fluorescence microscopy was used to measure the transfection effectiveness of EGFP in CHO cells (Figure 2A). Western blot analysis demonstrated that ERAP1 expression in transfected CHO cells was efficient (Figure 2B). Moreover, cells transfected with the ERAP1-001 variant produced a much greater amount of ERAP1 protein than those transfected with the *ERAP1*-002 variant (Figure 2C), corresponding with the results obtained from the PBMCs of homozygous *ERAP1*-001 and *ERAP1*-002 human individuals (Figure 1C). In addition, CHO cells expressing *ERAP1*-001 had considerably higher enzyme activity than *ERAP1*-002-expressing cells (*p* = 0.0105). (Figure 2D). Our research demonstrated that *ERAP1*-001 exhibited considerably more ERAP1 protein expression and enzyme activity than *ERAP1*-002.

### 3.6. ERAP1 Allelic Variants Affect the Crosstalk of ER Stress (UPR), Autophagy, and Inflammation

Recent evidence suggests that the axis of ER stress, autophagy, and inflammasome plays a crucial role in the etiology of inflammatory illness. U937 cells were transfected with *ERAP1* expression constructs in order to determine whether *ERAP1* variations impact the interaction between ER stress, autophagy, and inflammation. U937 cells transfected with *ERAP1*-001 displayed significantly higher levels of HLA-B27, ERAP1 protein, FHCs, FHC dimers, UPR markers (BiP, CHOP and XBP1), and inflammatory markers (Caspase 1 and IL-1) than U937 cells transfected with *ERAP1*-002. Moreover, U937 cells transfected with *ERAP1*-001 produced larger quantities of IL-23 in culture supernatants than U937 cells transfected with *ERAP1*-002. On the other hand, autophagy markers including Beclin 1 (I), LC3 I (J), LC3 II (K), caspase 1 (L), and IL-1β (M) were not significantly different between ERAP1-001 and ERAP1-002 cells.

## 4. Discussion

As a non-MHC susceptibility gene, *ERAP1* was found to interact with *HLA-B27* to increase the risk of AS [4,14]. The pathophysiological mechanism behind the impact of *ERAP1* allelic variations on AS susceptibility, on the other hand, is unknown. The common *ERAP1*-001 allelic variant was discovered to be a significant risk factor for AS in Taiwanese in the current investigation. According to our findings, epistasis between *ERAP1* allelic variants and *HLA-B* plays a crucial role in the development of AS in Taiwanese. *ERAP1*-001 induces a higher level of ERAP1 protein expression, enzyme activity, and IL-23 production than *ERAP1*-002. Increased ERAP1 enzyme activity appears to be a pathophysiological mechanism for the development of AS in *HLA-B27*^+^ individuals; therefore, inhibition of ERAP1 enzyme activity may be a viable treatment option for *HLA-B27*^+^ AS patients. In *HLA-B27*^−^ people, however, *ERAP1*-002 is a risk factor for AS. The importance of epistasis between *ERAP1* variants and *HLA-B27*^+^/ *HLA-B27*^−^ in the development of AS in Taiwanese is demonstrated.

ERAP1 is a multifunctional zinc-metallopeptidase that trims antigen peptides to the proper length for MHC-I molecules to display them. Antigenic epitopes are both produced and destroyed by ERAP1. The length of the peptidome is determined by interactions between the N-terminus of the antigen peptide and the enzyme active site of ERAP1, as well as the C-terminus of the peptide and an ERAP1 regulatory site. By changing the sequence and length of antigenic peptides deposited onto the corresponding MHC Class I molecules, ERAP1 affects a major portion of the immunopeptidome [20]. ERAP1 trims peptide precursors in solution to adjust the immunopeptidome’s optimal fit [21].

Peptidome studies of MHC-bound peptides in *HLA-B27* positive cell lines expressing various *ERAP1* variants reveal that *ERAP1* and *HLA-B27* have a functional relationship [22]. The structure, immunological effect, and peptide-trimming property of ERAP1 point to a mechanism or pathway for AS vulnerability that involves HLA-B27 [23,24]. ERAP1 directly affects peptide binding and presentation by HLA-B27, which could be a pathogenic mechanism for ERAP1’s participation in AS [25]. According to in vitro peptide catalysis experiments, *ERAP1* risk alleles with higher catalytic activity degrade HLA-B27 epitopes more efficiently, resulting in lower HLA-B27 presentation of the identical peptides [26].

mRNA stability, protein translation, and enzymatic activity may all be affected by combined effects of *ERAP1* genetic variations [27,28]. Between *ERAP*-001 (127R, 276I, 528K, and 730Q) and ERAP-002 (R127P, I276M, K528R, and Q730E), there are four residue differences. The molecular basis of ERAP1 SNVs on N-terminal peptidase function was discovered using crystal structure research [29]. The *ERAP1* SNV rs30187A > G (K528R) is found near the substrate pocket’s entrance, which may affect the C-terminus stability and aminopeptidase activity [29]. The R127P and K528R alterations affect ERAP1 conformational transition, whereas the Q730E change in ERAP1’s peptide-binding pocket may affect the peptides processed [30]. ERAP1 with 528R has lower enzymatic activity than ERAP1 with 528K, while ERAP1 with Q730E influences the lengths of antigenic peptides produced [22,25,28,31,32,33]. Therefore, the enzyme activity of ERAP1-002 (127P, 276M, 528R, and 730E) will be reduced. Furthermore, lower surface FHC expression on monocytes and HLA-B27-expressing APCs in AS patients was linked to 528R and 730E. HLA-B27 FHC surface expression was reduced by ERAP1 silencing or inhibition in APCs, as was IL-2 production by KIR3DL2CD3-reporter cells, and Th17 growth and IL-17A secretion by CD4+ T cells from AS patients [34].

In both white and East Asian ancestry populations, *ERAP1* has been genetically investigated in AS patients by sequencing [35] and intensive SNV mapping and imputation [4]. There were considerable differences in the presence and distribution of *ERAP1* coding SNVs among ethnic populations (Appendix A) [22,35,36,37,38,39]. The 1000 Genomes Project dataset discovered ten missense *ERAP1* SNVs, two of which are unusual in Taiwanese (residue positions 12 and 346) [38]. *ERAP1* SNV haplotypes, on the other hand, were comparable between the Asian 1000 Genomes Project population and Taiwanese (hap2 = *ERAP1*-001 plus -004, hap8 = *ERAP1*-002, hap7 = *ERAP1*, and hap10 = *ERAP1*-005, respectively). According to ERAP1 SNV data from the 1000 Genome Project, Reeves et al. detected a number of European-specific *ERAP1* SNVs using a cDNA cloning/sequencing technique [37], some of those SNVs may be produced by the cloning step. Based on five *ERAP1* SNVs, Robert et al. discovered three *ERAP1* haplotypes in the European population (349M/528K/575D/725R/730Q; 349M/528R/575D/725R/730E; 349V/528R/575N/725Q/730E). Those three *ERAP1* haplotypes could be deduced both in Europeans and Taiwanese (European 349M/528K/575D/725R/730Q = Taiwanese *ERAP1*-001 + 004, European 349M/528R/575D/725R/730E = Taiwanese *ERAP1*-002 + 003 + 006, and European 349V/528R/575N/725Q/730E = Taiwanese *ERAP1*-005). *ERAP1*-001 (56E, **127R**, **276I**, 349M, **528K**, 575D, 725R, and **730Q**) and *ERAP1*-002 (56E, **127P**, **276M**, 349M, **528R**, 575D, 725R, and **730E**) differ at four residues (R127P, I276M, K528R, and Q730E) and two intron SNVs. Two intron SNVs are thought to alter ERAP1 functions [35,40]. Importantly, we examined 11 *ERAP1* SNVs and identified two common allelic variants (haplotypes) in Taiwanese. To the best of our knowledge, for the first time in human subjects, we looked at the functional differences between the two most frequent ERAP1 variations in depth, which differed from previous functional investigations of *ERAP1* variants [22,37,40,41].

Reduced ERAP1 mRNA expression in the presence of a coding variant has been linked to Behçet’s Disease (BD) [42]. With longer epitopes, the promiscuous sub-peptidome binding property of HLA-B51 indicates a genetic effect on dysregulated CD8+ T response and aberrant NK cell activation, leading to the development of BD [43]. Other *HLA-B* alleles have been involved in AS development in European populations [6], therefore ERAP1 epistasis may not be limited to *HLA-B27*. *HLA-B40* increased susceptibility to AS in *HLA-B27*^−^ AS patients in Taiwanese [44], but not in Caucasian [45]. As shown in Appendix A, *HLA-B40:0*1 frequencies (phenotype frequency = 37.1%, allele frequency = 18.55%) among 62 *HLA-B27*^−^ AS patients were similar to those in general Taiwanese population, indicating that *HLA-B40* may not play a unique role in AS. *HLA-B27*^−^ Taiwanese AS patients have lower ERAP1 enzyme activity, suggesting that decreased ERAP1 enzyme activity may play a role in AS. Understanding the pathophysiology of AS in people will require more research into the link between *HLA-B* alleles and *ERAP1* variations. 

Misfolding of HLA-B27 HC promotes ER stress and activation of the UPR. BiP and XBP1 are two UPR indicators that are increased upon the activation of the three major UPR pathways. Compared to cells expressing the *ERAP1*-002 variant, cells expressing the *ERAP1*-001 variant produced larger quantities of ERAP1 protein, FHCs, FHC dimers, BiP, XBP1, and IL23. The increased synthesis of BiP and XBP1 in cells expressing *ERAP1*-001 suggests that the overexpression of MHC class I FHCs and FHC dimers results in the activation of UPR, the elevation of IL-23 levels, and proinflammatory response. The enhanced expression of FHCs and FHC dimers in cells bearing HLA-B27 may explain why the link between *ERAP1*-001 and AS susceptibility is exclusive to *HLA-B27*^+^ people.

Autophagy is a lysosome-dependent degradation mechanism that destroys misfolded proteins and damaged ER. UPR not only enhances cytokine-mediated inflammatory responses leading to disease, but also promotes autophagy. The control of autophagy by reducing ER stress could therefore provide cytoprotection [46]. PBMCs expressing the *ERAP1*-001 variant displayed significantly higher levels of autophagy markers (Beclin 1, LC3 I, and LC3 II), inflammation markers (caspase 1 and IL-1), and IL-23 (p19 and p40) compared to PBMCs expressing the *ERAP1*-002 variant (Figure 1). Increased productions of BiP, CHOP, and XBP1 in cells expressing *ERAP1*-001 suggest that *ERAP1*-001 may cause the misfolding of HLA-B complexes or HLA-B27 associated with abnormal peptides, resulting in the activation of the UPR [47] and subsequent induction of autophagy (Beclin 1, LC3 I and LC3 II), inflammation (caspase 1 and IL-1β) [48]. Nonetheless, U937 cells transfected with *ERAP1*-001 displayed comparable levels of autophagy markers (Beclin 1, LC3 I and LC3 II) compared to those with *ERAP1*-002 (Figure 3). In this regard, we speculated that endogenous ERAP1 may mask the effect of *ERAP1*-001 and *ERAP1*-002 variants on autophagy in U937 cells.

Although the exact mechanism underlying IL-23 overexpression remains unknown, IL-23 was detectable in the peripheral blood and tissues of AS patients. In particular, it is unknown if the overproduction of IL-23 could be linked to the misfolding of HLA-B27 by inducing the UPR [49]. On the other hand, the proof that IL-23 is active at the surface of the intestinal mucosa [50] and the evidence that certain types of bacterial stimuli may alter its intestinal expression [51] suggest the role of microbes in the IL-23 overexpression observed in AS [52]. Multiple roles could be carried out by ERAP1 in the extracellular area [53]. ERAP1 produced into the circulation in response to viral or inflammatory stimuli that boost NO production may function as an acute-phase host defense protein [54]. Exosomes contain ERAP1, TNF-α, IFN-γ, and CCL3, which play a crucial role in the inflammatory process by activating macrophages [55]. The immunomodulation cytokine profiles of the IL-17/IL-23 pathway are affected by the *ERAP1* SNPs rs30187, rs2287987 [56], and rs27038 [57]. Emerging immunological and genetic evidence validated the central function of the IL-17A/IL-23 axis, resulting in the continued development of new medications targeting distinct components, such as medicines blocking the p40 subunit of IL-12/IL-23 or the p19 subunit of IL-23 for therapeutic purposes. Our results indicate that the genetic relationship between HLA-B27 and ERAP1 is a critical predictor of IL-23 pathogenesis in AS.

## 5. Conclusions

The present investigation definitively determined a functional role of *ERAP1* allelic variants In AS. Ex vivo and in vitro research demonstrated that *ERAP1* allelic variations have a substantial effect on ERAP1 expression and enzyme activity, FHCs, FHC dimers, and UPR markers (BiP, CHOP and XBP1). Two common ERAP1-001 and ERAP1-002 variants are associated with susceptibility to AS in Taiwanese in an HLA-B27-dependent manner.

## Figures and Tables

**Figure 1 cells-11-02427-f001:**
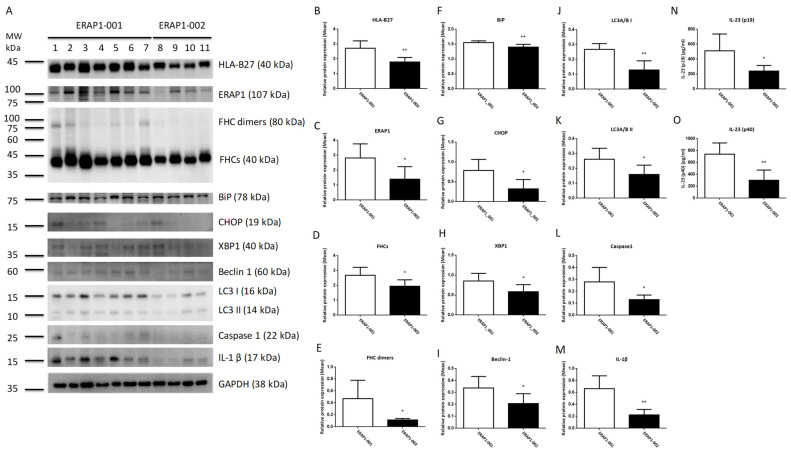
PBMCs from ERAP1-001 and ERAP1-002 homozygous human subjects produced significantly different levels of ERAP1 and related proteins. Proteins in cell lysates of PBMCs were detected by Western blot analysis and a panel of proteins from seven ERAP1-001 and four ERAP1-002 homozygous donors were analyzed (**A**). Band intensities were quantified from the digital image by densitometry using Image and were normalized to the loading control. PBMCs of ERAP1-001 homozygous donors produced significantly higher levels of HLA-B27 (**B**), ERAP1 (**C**), MHC class I free heavy chain or FHCs (**D**), FHC dimers (**E**), immunoglobulin heavy-chain binding protein (BiP) (**F**), CCAAT-enhancer-binding protein homologous protein (CHOP) (**G**), X-box-binding protein 1 (XBP1) (**H**), Beclin-1 (**I**), LC3 I (**J**), LC3 II) (**K**), Caspase 1 (**L**), and IL-1β (**M**) as compared to ERAP1-002 homozygous donors. ELISA was used to detect IL-23 (p19/p40) in culture media. PBMCs from ERAP1-001 homozygous donors produced significantly higher level of IL-23 p19 (512.1 ± 84.94 versus 241.9 ± 36.40, *p* = 0.0478) (**N**) and p40 (740.4 ± 71.25 versus 301.7 ± 85.72, *p* = 0.0041) (**O**) than PBMCs of ERAP1-002 homozygous donors (data are shown as means with SD). Data are representative of three experiments. * *p* < 0.05, ** *p* < 0.01.

**Figure 2 cells-11-02427-f002:**
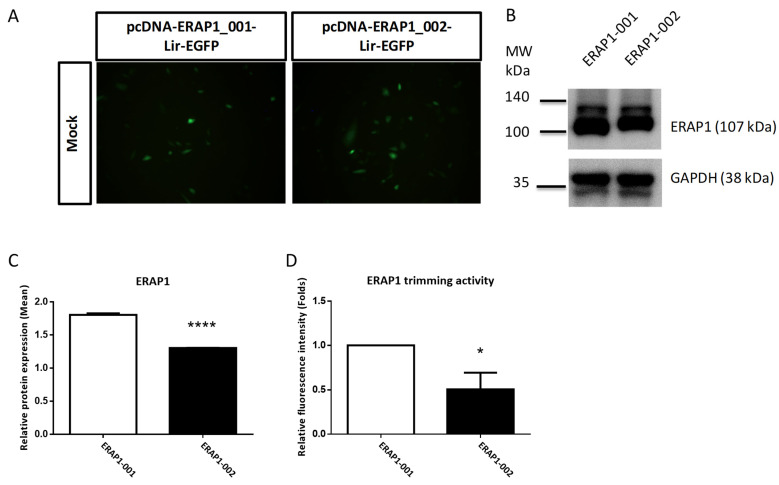
*ERAP1*-001 exhibited considerably more ERAP1 protein expression and enzyme activity than *ERAP1*-002 in transfected cells. (**A**) Representative images of CHO cells transfected with pcDNA-ERAP1-Lir-EGFP expression constructs. All pictures were taken with a FITC channel (Green) with a 450/490-nm filter set with the same exposure time (900 ms). (**B**,**C**) Western blot analysis of cellular lysates from transfected U937 cells with pcDNA-ERAP1-EGFP. Band intensities were quantified and shown as normalized to the loading control. (**D**) ERAP1 enzyme activity analysis. Enzymatic reactions were performed via a continuous florigenic assay. Data are one representative of three experiments. Data are shown as mean (SD). * *p* < 0.05; **** *p* < 0.0001.

**Figure 3 cells-11-02427-f003:**
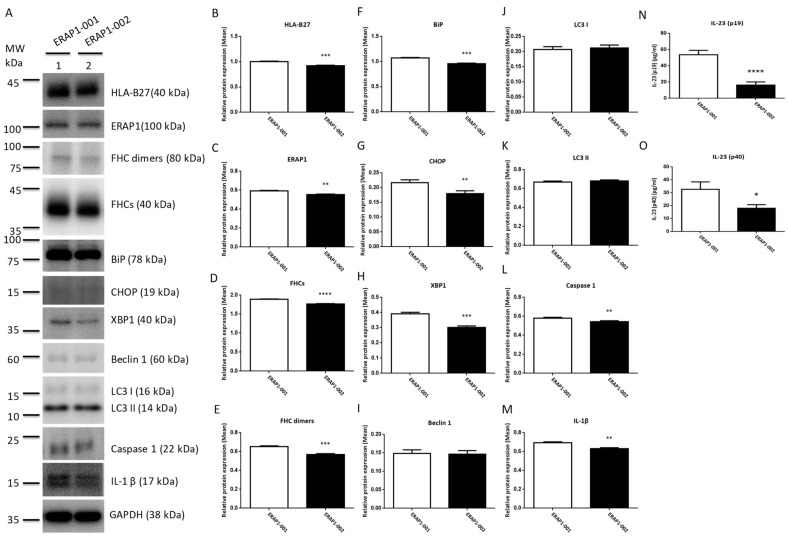
Effects of ERAP1-001 and ERAP1-002 variants on misfolding proteins, UPR markers, autophagy markers, inflammation markers, and IL-23 in transfected U937 cells. Proteins in cell lysates of transfected U937 cells were detected by Western blot analysis and a panel of proteins from were analyzed (**A**). GAPDH was used as the loading control where indicated. Band intensities were quantified and normalized to the loading control. Protein levels of HLA-B27 (**B**), ERAP1 (**C**), FHCs (**D**), FHC dimers (**E**), BiP (**F**), CHOP (**G**), XBP1 (**H**), caspase 1 (**L**), and IL-1β (**M**) were significantly higher in ERAP1-001 U937 cells as compared to ERAP1-002 cells. Protein levels of Beclin 1 (**I**), LC3 I (**J**), LC3 II (**K**), caspase 1 (**L**), and IL-1β (**M**) were not significantly different between ERAP1-001 and ERAP1-002 cells. Culture media were collected for ELISA to detect IL-23 -p19/p40. U937 cells transfected with *ERAP1*-001 produced larger quantities of IL-23-p19 subunit (*p* < 0.0001, 53.78 ± 2.696 versus 16.28 ± 1.964) (**N**) and IL-23-p40 subunit (*p* = 0.0158, 32.74 ± 3.288 versus 18.10 ± 1.563) (**O**) than U937 cells transfected with *ERAP1*-002 (mean ± SD are shown). Combined data were from three experiments. * *p* < 0.05; ** *p* < 0.01, *** *p* < 0.001, and **** *p* ≤ 0.0001.

**Table 1 cells-11-02427-t001:** Association of *ERAP1* allelic variants with AS susceptibility in Taiwanese.

*ERAP1* Allelic Variant *	Estimated Frequency			Permutation **	Logistic Regression		Logistic Regression Adjusted for Sex	
	AS	Normal	All	*p* Value	*p* Value	OR (95% CI)	*p* Value	OR (95% CI)
	(2N = 1726)	(2N = 2876)	(2N = 4602)					
001	46.85%	36.69%	40.50%	<0.000001	1.32 × 10^−11^	1.53 (1.35–1.73)	1.39 × 10^−11^	1.53 (1.35–1.73)
002	21.91%	23.41%	22.85%	0.23895	0.2390	0.92 (0.79–1.06)	0.2419	0.92 (0.79–1.06)
003	11.91%	12.43%	12.23%	0.5833	0.5905	0.95 (0.79–1.15)	0.5874	0.95 (0.79–1.15)
004	7.79%	6.96%	7.27%	0.2902	0.2938	1.13 (0.90–1.42)	0.2952	1.13 (0.90–1.42)
005	3.63%	3.62%	3.63%	0.99285	0.9898	1.00 (0.72–1.39)	0.9917	1.00 (0.72–1.39)
006	2.08%	2.41%	2.29%	0.4558	0.4541	0.85 (0.56–1.29)	0.4531	0.85 (0.56–1.29)

* Eleven SNVs including rs3734016, rs26653, rs26618, rs2287987, rs30187, rs10050860, rs469783, rs17482078, rs27044, rs27980, and rs27037 were used to determine *ERAP1* allelic variants as shown in the Appendix A. ** *p*-values for *ERAP1* variants were generated using the expectation-maximization (EM) algorithm with 10,000 permutations.

**Table 2 cells-11-02427-t002:** Association of *ERAP1* allelic variants with HLA-B27 positivity among Taiwanese AS patients stratified by HLA-B27 status.

*ERAP1* Allelic Variant *	Estimated Frequency			Permutation **	Logistic Regression		Logistic Regression Adjusted for Sex	
	HLA-B27^+^	HLA-B27^−^	All	*p* Value	*p* Value	OR (95% CI)	*p* Value	OR (95% CI)
	(2N = 1602)	(2N = 124)	(2N = 1726)					
001	47.77%	36.80%	46.93%	0.0118	0.0132	1.62 (1.11–2.36)	0.0147	1.61 (1.10–2.36)
002	20.89%	33.95%	21.89%	2.00 × 10^−4^	0.0005	0.50 (0.34–0.74)	0.00064	0.50 (0.34–0.75)
003	12.07%	10.90%	11.98%	0.674	0.6776	1.13 (0.63–2.05)	0.7637	1.10 (0.60–2.00)
004	8.18%	3.10%	7.80%	0.0293	0.0458	2.82 (1.02–7.79)	0.0409	2.90 (1.04–8.04)
005	3.54%	5.12%	3.66%	0.3132	0.3408	0.66 (0.28–1.54)	0.3741	0.68 (0.29–1.60)
006	2.10%	1.84%	2.08%	0.8378	0.8398	1.15 (0.30–4.40)	0.8863	1.10 (0.29–4.27)

* Eleven SNVs including rs3734016, rs26653, rs26618, rs2287987, rs30187, rs10050860, rs469783, rs17482078, rs27044, rs27980, and rs27037 were used to determine *ERAP1* variants as shown in the Appendix A. ** *p*-values for *ERAP1* variants were generated using the expectation-maximization (EM) algorithm with 10,000 permutations.

**Table 3 cells-11-02427-t003:** Distributions of *ERAP1* allelic variants in HLA-B27^−^ AS patients (AS B27^−^) and HLA-B27^−^ healthy controls (Normal B27^−^).

*ERAP1* Allelic Variant *	Estimated Frequency			Permutation **	Logistic Regression		Logistic Regression Adjusted for Sex	
	AS B27^−^	Normal B27^−^	All	*p* Value	*p* Value	OR (95% CI)	*p* Value	OR (95% CI)
	(2N = 124)	(2N = 2686)	(2N = 2810)					
001	36.64%	36.33%	36.35%	0.9453	0.9448	1.01 (0.70–1.47)	0.9548	1.01 (0.70–1.47)
002	34.53%	23.65%	24.13%	0.0045	0.0058	1.72 (1.17–2.52)	0.0080	1.69 (1.15–2.49)
003	11.63%	13.11%	13.04%	0.6147	0.6205	0.86 (0.48–1.54)	0.6614	0.88 (0.49–1.57)
004	5.84%	5.40%	5.42%	0.8202	0.8245	1.10 (0.49–2.48)	0.8515	1.08 (0.48–2.45)
005	3.13%	6.80%	6.64%	0.0966	0.1174	0.43 (0.15–1.24)	0.1075	0.42 (0.15–1.21)
006	2.75%	1.98%	2.01%	0.5287	0.5242	1.46 (0.46–4.67)	0.5207	1.46 (0.46–4.67)

* Eleven SNVs including rs3734016, rs26653, rs26618, rs2287987, rs30187, rs10050860, rs469783, rs17482078, rs27044, rs27980, and rs27037 were used to determine *ERAP1* variants as shown in the Appendix A. ** *p*-values for *ERAP1* variants were generated using the expectation-maximization (EM) algorithm with 10,000 permutations.

## Data Availability

Not applicable.

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
