# Peer review of "Functional ERAP1 Variants Distinctively Associate with Ankylosing Spondylitis Susceptibility under the Influence of HLA-B27 in Taiwanese"

_cells, 2022, doi:10.3390/cells11152427_

Round 1
Reviewer 1 Report
Section materials and methods – paragraph „Statistical analysis” should be at the end of Section.
Please add information on how 45 patients were selected for sequencing.
Tables 1, 2, and 3 should be moved to the right paragraphs.
Why these patients (……were extracted from 11 human participants (seven ERAP1-001 homozygous AS patients and four ERAP1-002 AS patients……) were selected?
Figures 2 and 3 should be placed in the result section
Author Response
Reviewer #1:
Major comments:
- Section materials and methods – paragraph “Statistical analysis” should be at the end of Section.
Response: The “Statistical analysis” paragraph was rearranged to the end of Materials and Methods section according to the reviewer’s comment.
- Please add information on how 45 patients were selected for sequencing.
Response: We added the information about the selection of 45 human subjects for discovery purpose on page 5 of the revised manuscript.
- Tables 1, 2, and 3 should be moved to the right paragraphs.
Response: Table 1, 2, and 3 were moved to the correct places in the revised manuscript.
- Why these patients (were extracted from 11 human participants (seven ERAP1-001 homozygous AS patients and four ERAP1-002 AS patients) were selected?
Response: We added the reasons for the selection of seven ERAP1-001 homozygous AS patients and four ERAP1-002 AS patients to examine the epistaxis effects if ERAP1 haplotypes influence phenotypes in HLA-B27 positive individuals
- Figures 2 and 3 should be placed in the result section.
Response: Figure 2 and 3 were moved to the correct places in the revised manuscript.
Reviewer 2 Report
I've read the article "Functional ERAP1 variants distinctively associate with ankylosing spondylitis susceptibility under the influence of HLAB27 in Taiwanese" with interest. It represents an important study aimed to examine how prevalent ERAP1 allelic variants (SNV haplotypes) in Taiwan affect ERAP1 functions and AS susceptibility in the presence or absence of HLA-B27. The experiment design and the presentation of data are excellent. I congratulate the authors. However, several points should be addressed by the authors.
Comments:
1. The authors should be provided the information for all the regents in the MM section like this (Company name, city, country).
2. For all or most techniques that have been described in the MM section, it is suggested to use a relevant reference.
3. Figure legends are highly incomplete especially figure legends 1 and 3 as well as supplementary figures. The authors have written the figure legends like the headings of the results section and this way is not correct. The authors needed to write a proper title related to the figures and then describe the results of each section briefly and separately (A to…….).
4. I am not satisfied with the conclusion section. The authors should improve this section.
Author Response
Reviewer #2:
Major comments:
- The authors should be provided the information for all the regents in the MM section like this (Company name, city, country).
Response: Company name, city, and country for all reagents from vendors were provided in the revised manuscript.
- For all or most techniques that have been described in the MM section, it is suggested to use a relevant reference.
Response: Most of methods and materials in this study were described in details and could be easily replicated by other investigators. As for the unique technique of ERAP1 enzyme activity determination that could be unfamiliar to audience, we cited the reference # 17 on the page 6 in the Materials and Methods section.
- Figure legends are highly incomplete especially figure legends 1 and 3 as well as supplementary figures. The authors have written the figure legends like the headings of the results section and this way is not correct. The authors needed to write a proper title related to the figures and then describe the results of each section briefly and separately (A to…….).
Response: We revised figure legends of Figure 1 and 3 as advised by the reviewer.
- I am not satisfied with the conclusion section. The authors should improve this section.
Response: We revised the conclusion section.